# Quantitative characterization of recombinase-based digitizer circuits enables predictable amplification of biological signals

Katherine A. Kiwimagi[1,6], Justin H. Letendre[2,6], Benjamin H. Weinberg [2], Junmin Wang [3], Mingzhe Chen[1], Leandro Watanabe[4], Chris J. Myers [4], Jacob Beal[5✉], Wilson W. Wong [2✉] & Ron Weiss [1✉]

Many synthetic gene circuits are restricted to single-use applications or require iterative refinement for incorporation into complex systems. One example is the recombinase-based digitizer circuit, which has been used to improve weak or leaky biological signals. Here we present a workflow to quantitatively define digitizer performance and predict responses to different input signals. Using a combination of signal-to-noise ratio (SNR), area under a receiver operating characteristic curve (AUC), and fold change (FC), we evaluate three small-molecule inducible digitizer designs demonstrating FC up to 508x and SNR up to 3.77 dB. To study their behavior further and improve modularity, we develop a mixed phenotypic/mechanistic model capable of predicting digitizer configurations that amplify a synNotch cell-to-cell communication signal (Δ SNR up to 2.8 dB). We hope the metrics and modeling approaches here will facilitate incorporation of these digitizers into other systems while providing an improved workflow for gene circuit characterization.

[1] Biological Engineering, Massachusetts Institute of Technology, Cambridge, MA, USA. [2] Department of Biomedical Engineering and Biological Design Center, Boston University, Boston, MA, USA. [3] The Bioinformatics Graduate Program, Boston University, Boston, MA, USA. [4] Department of Electrical and Computer Engineering, University of Utah, Salt Lake City, UT, USA. [5] Raytheon BBN Technologies, Cambridge, MA, USA. [6] These authors contributed equally: Katherine A. Kiwimagi, Justin H. Letendre. ✉email: jake.beal@raytheon.com; wilwong@bu.edu; rweiss@mit.edu

Synthetic biology aims to solve a wide range of problems through the construction of biological parts and their assembly into systems. In particular, engineered genetic circuits have shown promise for improving therapeutics[1,2], disease diagnosis[3], and metabolic engineering[4]. Understanding the limits and design rules for programming these circuits to execute a particular biological function is essential for implementing more advanced synthetic systems. As systems of synthetic circuits become more complex, one of the most fundamental challenges in their design is the ability to maintain desired signal features, such as the distinction between different cell states and the reproducibility of protein and other output levels from circuits in different contexts. An underlying cause of these challenges is the lack of full modularity of circuit parts; parts are often characterized in specific contexts which fail to describe their full range of activity, limiting the exchange of these components between different systems such as when connected to unique input systems (small molecule, synthetic receptors, light, etc.). Using standard metrics to evaluate circuit performance and identifying important biological mechanisms that govern their activity is critical for accurately comparing and predicting circuit responses, and for creating truly modular components to tune their activity.

Recombinase-based analog-to-digital converters (digitizers), which transform a graded analog response into a near digital Boolean one, have been previously explored in mammalian cells to improve the quality of biological signals[5,6]. While analog signals allow cells to integrate and respond to environmental cues in a reversible, dose-dependent manner, digital signals are essential for governing key decisions in cells such as differentiating toward a particular lineage during development or inducing apoptosis. Given the power of these digitizers in their intended settings, it will be useful to probe their design rules to generalize digitizer designs for use in cell-based signal processing. Without an in depth understanding of the components and mechanisms responsible for their performance, implementing digitizers in new settings requires extensive ad hoc changes to these systems.

Site-specific recombinases, such as Cre and Flp, are proven components for use in genetic circuits and are particularly well suited for use in digitizer designs due to their all-or-nothing behavior in recombining segments of DNA between target sites[6–10]. In addition, the large array of orthogonal inducible recombinase systems[11–13] endows them with the versatility to be used with multiple inputs systems. Many implementations of recombinase technology, however, demonstrate low tolerance to leaky input signals that can lead to unintended component expression, recombination, and deterioration of device performance[14]. While previous work has demonstrated that feedforward RNAi-mediated control of recombinase expression can reduce this leaky behavior to levels below detectable limits[5,15], comparing different circuit architectures should provide further insight into the limits of the digitizer design and elucidate areas of divergence in their fundamental behavior for use in different applications. To meet this need, here we present and compare three designs: (1) no-shRNA, (2) feedforward-shRNA, and (3) constant-shRNA mediated recombinase-based digitizers.

While defining the performance of our circuits, we find that there is no set of established, standardized metrics for quantifying common genetic circuit attributes. One of the most common metrics is fold change, the mean ON-state expression level divided by the mean OFF-state expression level of a circuit output gene[16,17]. However, fold change does not describe variance in the cell populations producing these signals and therefore may not be an ideal metric to assess signal quality, the distinguishability between signal states. To improve our characterization framework we use signal-to-noise ratio (SNR), which captures both signal amplitude and variance and is frequently used by many other engineering disciplines, to quantify signal performance. This is further supplemented by area under the curve (AUC) of a receiver operating characteristic (ROC) curve, another distribution based metric to capture distinguishability. We further develop a mathematical model of the digitizers to determine the governing biological mechanisms contributing to each signal feature. We validate our model by predicting circuit performance for each digitizer topology to improve our understanding of the most fundamental parameters governing their performance and modularity. To demonstrate the power of these circuits coupled with our characterization efforts and predictive model (Fig. 1A), we rationally compose each digitizer together with a synthetic notch (synNotch) sensor to create an enhanced cell-to-cell communication switch, demonstrating signal amplification among digitizer-expressing receiver cells in the presence of ligand-expressing sender cells.

## Results

**Design of digitizer.** To create a digitizer, we first assembled a drug-inducible recombinase module. Site-specific recombinases offer the advantage of inherent memory by cleaving and recombining specific segments of DNA, leading to a near-permanent change in DNA state[18]. The Tet-ON system, using the doxycycline (dox) inducible rtTA-Advanced transcription factor and target TRE-tight promoter (pTRE), has proven to exhibit tight control of gene expression and has been previously used for inducible Cre recombination[12,19]. As a proof of concept for our digitizers, we placed a codon optimized flippase (Flp) tyrosine recombinase under the control of the pTRE (Fig. 1B). In the presence of dox, rtTA-Advanced should bind to the pTRE and induce analog expression of Flp. Recombinase activity is reported through the excision of a transcription termination sequence flanked by Flp recognition target (frt) sites placed between a CAG promoter and a GFP coding sequence (frt-STOP-frt-GFP, FSF-GFP), acting as the digital ON or OFF signal.

We generated a dox dose response curve for this digitizer module to determine the GFP output fluorescent protein (OFP) response to several levels of dox input at a fixed time scale (Supplementary Fig. S1). We also determined the level of leaky expression from the pTRE promoter and how this affects system performance over a period of 4 days (time series, Supplementary Fig. S2). For the dose-response experiments, HEK293FT cells were transfected with the digitizer and immediately induced by titration of a dox stock solution to the media (see Methods). Flow cytometry data was collected 48 h post induction and a titration curve was generated based on the geometric mean of the top 30% of single cells expressing a constitutive fluorescent protein (CFP) in each population. We find that other binned transfection groups maintain the same qualitative trends (Supplementary Fig. S3), and choose to focus on the top 30% of CFP expressing cells to standardize the cell populations we compare. Using this design, OFP activity reaches saturation at a dox concentration of 14 nM and displays an eightfold higher output expression when induced compared with basal uninduced levels. However, we note that there is a significant population of cells generating an OFP signal in the uninduced OFF-state, indicated by a large shoulder in the OFP distributions (Supplementary Fig. S1). This leaky behavior is thought to be due to basal Flp expression from the pTRE. Time series characterization over four days post transfection/induction of the digitizer supports this hypothesis, showing a gradual increase in aberrant recombination over time as a steadily growing OFP+ shoulder population of uninduced cells (Supplementary Fig. S2A). This caused the device's fold change to peak at 48 h post induction followed by a decrease in performance (Supplementary Fig. S2B) that undermines the device's capacity

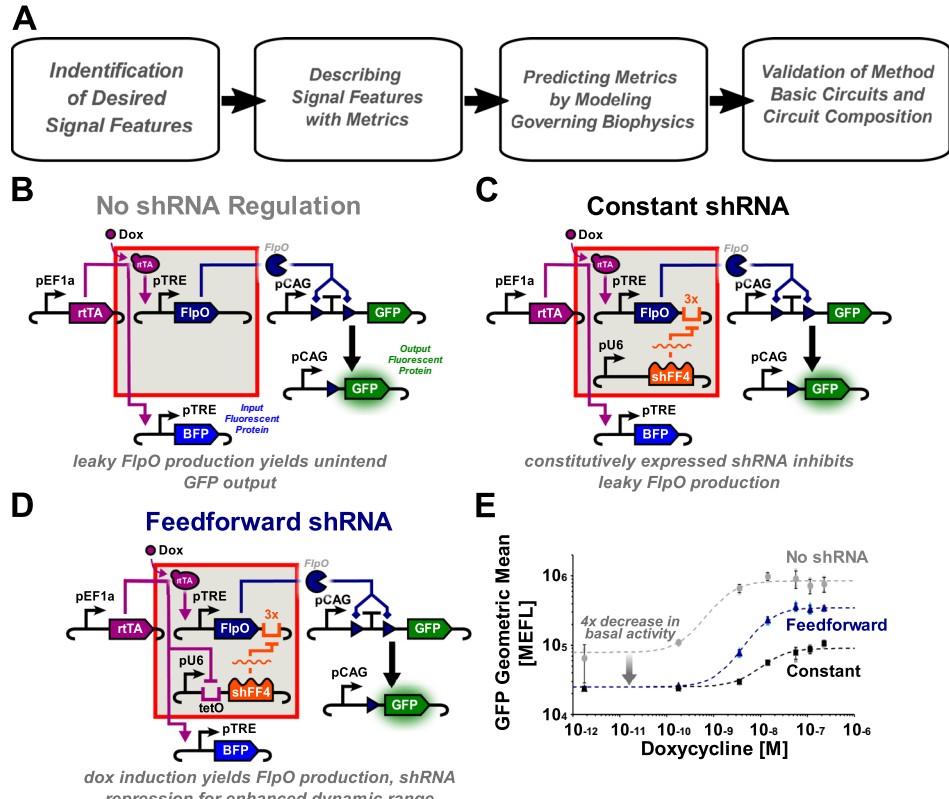

**Fig. 1 Characterization flow chart and digitizer designs. A** Flow chart representing the quantitative standardization demonstrated in this paper.
**B** Recombinase digitizer circuit design connecting a dox-inducible Flp recombinase with an FSF-GFP output and incorporating no-shRNA regulation.
**C** Recombinase digitizer circuit design incorporating constantly expressed shRNA. **D** Recombinase digitizer circuit design incorporating feedforward
regulated shRNA whose expression can be turned off through the induction of rtTA binding to a tet operator (tetO) site. **E** shRNA serves to mitigate leaky
expression, feedforward circuit topology shows enhanced dynamic range under a constrained Flp:shRNA ratio compared with constitutively expressed
shFF4. Doxycycline dose response profiles of all three recombinase amplifier circuits transiently transfected into HEK293FT cells ($n = 3$), error bars
represent SEM.

for both digital responses and memory of an input signal, necessitating greater control over Flp expression.

Previous work[5] has shown that incorporating an shRNA element controlled by a coherent feedforward loop is effective in stifling leaky recombinase expression. To control leaky Flp expression in our circuit's OFF-state, we present two additional designs incorporating constant- (Fig. 1C) and tet-repressible coherent feedforward-shRNA (Fig. 1D). Each shRNA element was added to the circuit as a separate plasmid, and Flp was tagged with three shRNA-FF4[11] target sites on its 3′ UTR to promote efficient repression. For the constant-shRNA topology, shRNA is produced at a steady level establishing a transcriptional threshold that the pTRE must overcome to yield Flp protein. The feedforward-shRNA topology includes a TET operator site between the hU6 promoter and shRNA transcript; using this binding site, shRNA transcription is stifled by the addition of dox and binding of rtTA to the operator site while simultaneously promoting Flp transcription from the pTRE. By testing both constant- and feedforward-shRNA elements, we considered the differences in the regulatory capabilities of each circuit. In addition, we chose to incorporate each transcriptional unit as a separate plasmid to allow for tuning of each circuit element in response to variable levels of basal Flp expression. Dose-response characterization of these new constant and feedforward shRNA circuit designs at a Flp:shRNA plasmid ratio of 1:10 revealed a dramatic (fourfold) decrease in basal OFP expression at 48 h post induction (Fig. 1E, Supplementary Fig. S2). While the maximum

level of induction (OFP signal) was reduced in both circuits incorporating shRNA, the fold change in OFP increased for the feedforward circuit (15-fold) as compared to the no-shRNA design (8.5-fold) at 48 h when induced with 225 nM dox. While the fold change of the constant shRNA circuit is very low (4.5-fold), this can be attributed to the low level of activation caused by over-repression of Flp transcripts. Many cells fail to turn ON even at the highest dox level, with the ON-state visible as a small shoulder instead of a robust population-level response (Supplementary Fig. S2A). However, both the constant and feedforward shRNA circuits are able to control leaky expression more effectively than the original design. Time series characterization of all three circuit topologies confirms a dramatic decrease in leaky OFF state Flp expression for both shRNA systems at all time points (Supplementary Fig. S2B).

**Metrics for functional characterization of digitizer modules.** There does not currently exists a single, standard set of metrics to describe circuit performance among existing literature. While fold change, the amplitude between mean population expression levels, is one of the most common metrics used to describe the performance of biological circuits and is useful for describing signal amplification, it is not sufficient for describing distinguishability of population states or tolerance to leaky component expression when no input is provided. We therefore choose to apply additional metrics to provide a more complete basis for circuit characterization.

**Box 1 | AUC Explained**

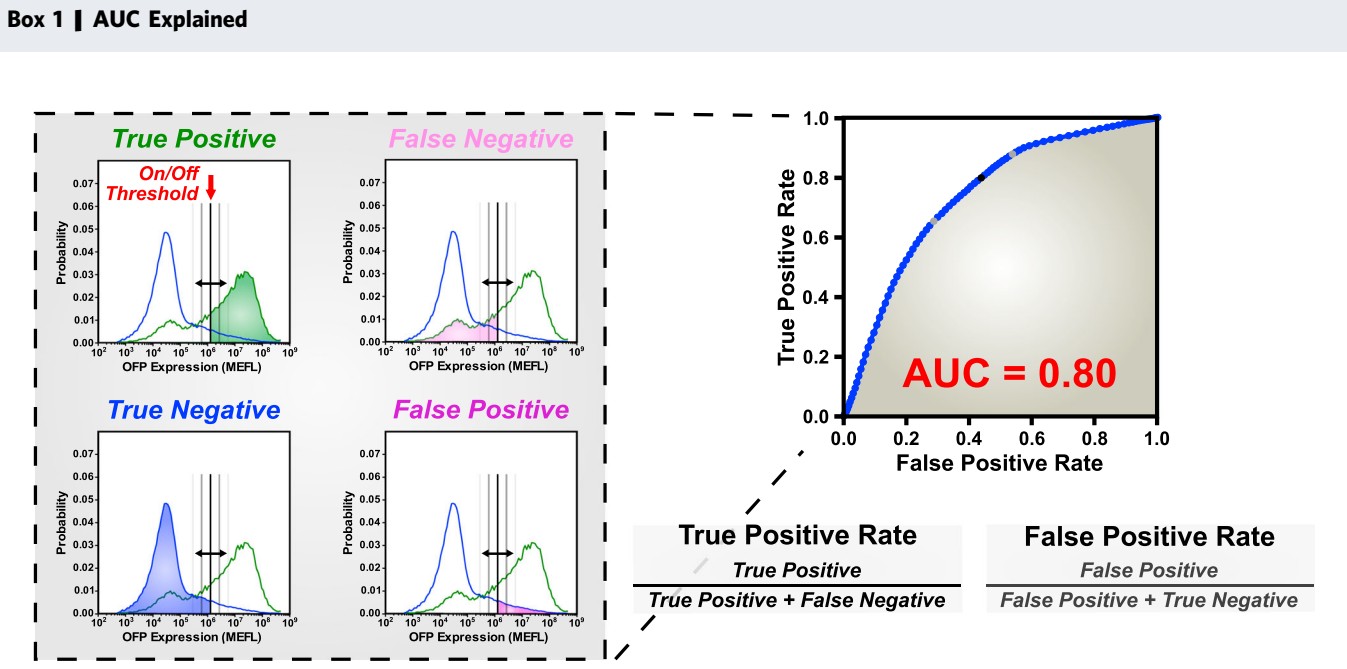

**Definition:** Area under the curve is a measure of how distinguishable two signals are from one another. In this case, populations of cells expressing fluorescent proteins are compared and AUC is the calculated area underneath a receiver operator characteristic (ROC) curve comparing the false negative vs. true positive rate for fluorescent protein expression. To generate the ROC from flow cytometry data as we have done here, each point is plotted using unique, arbitrary threshold values differentiating ON- and OFF-state cells. These threshold values are varied across the entire range of fluorescent protein expression data from both populations combined (shown at left), and cells from each population are then identified as falling above (ON) or below (OFF) the given threshold expression value. Given the number of ON- and OFF-state cells in each population for each threshold value, four percentages are calculated: false positive, false negative, true positive and true negative (left). False positive rate vs. true positive rate are then calculated and plotted for each arbitrary threshold value, and the AUC can be determined from the generated ROC curve (right). It is important to note that when AUC = 1 the positive and negative distributions are completely distinguishable, while an AUC = 0.5 indicates that populations are indistinguishable form one another.

**Biological relevance:** AUC gives a measure of distigushability that is independent of knowing an exact threshold value for ON and OFF cells.

**History:** AUC has been widely used in drug discovery assays as well as electronic circuit design.

We define signal quality as the ability to distinguish between the ON- and OFF-states of a population of cells expressing a given circuit. The area under a receiver operating characteristic (ROC) curve (AUC) has been used as a measure of distinguishability for many diagnostic devices and biological assays. AUC can easily be calculated from flow cytometry data of the ON- and OFF-state cell populations (Box 1), and can be used as a measure of distinguishability[20,21].

While AUC is straightforward to calculate and may be superior to fold change in measuring distinguishability, it does not capture the magnitude of each signal or the amplitude between ON- and OFF-states. We are interested in increasing this amplitude while minimizing the variance of each state, creating a more stable and reliable circuit. Signal-to-noise ratio (SNR) supplements fold change in its ability to capture both variance and signal strength. Previous work has shown that the SNR of unimodal biological signals with symmetric variation can be estimated by a square wave signals approximation whose SNR can be calculated using both the first and second mean centered moments of the OFF- and ON-state distributions[22]. For our digitizers, however, false positive and false negative cell states produce a bi-modal distribution and variation may not be symmetric, as depicted in Box 2. Therefore, we extend the SNR calculation to also include the power of the noise due to false positives and false negative.

To apply these metrics to our data set, we quantify input and output signal levels using fluorescent proteins as system readouts

(Fig. 1B, Supplementary Fig. S3). We denote these proxy markers as the input fluorescent protein (IFP), which reports on the transcriptional activity of the inducible pTRE producing Flp to measure the input signal strength, and output fluorescent protein (OFP), indicating the activity of Flp protein (Supplementary Fig. S3). In addition to the IFP and OFP readouts from our device, we also incorporate a constitutive fluorescent protein (CFP) to analyze specific cell populations based on transfection level which can be correlated with plasmid copy number (Supplementary Fig. S3)[23]. There does not appear to be a significant qualitative difference in the behavior of cells transfected with lower copy numbers of circuit plasmids, as defined by CFP (Supplementary Fig. S4). We thus choose to isolate and analyze the top 30% of CFP expressing cells in each population to compare across cells transfected with similar amounts of DNA for all data presented in this work.

When we evaluate each system at a fixed recombinase:shRNA ratio, we note that all three digitizer topologies perform distinctly different from one another over time. The no-shRNA topology (Fig. 1B) exhibits substantial leakiness of OFF state cells transitioning into the ON (Supplementary Fig. S2B), while basal Flp expression in the constant shRNA topology (Fig. 1C) is mitigated completely up to 96 h. However, at the 1:10 Flp:shRNA ratio tested, the shRNA repression at a Flp:shRNA plasmid weight ratio of 1:10 provides a transcriptional threshold too great for the dox induction to overcome and this circuit fails to achieve

**Box 2 ▌ SNR Explained**

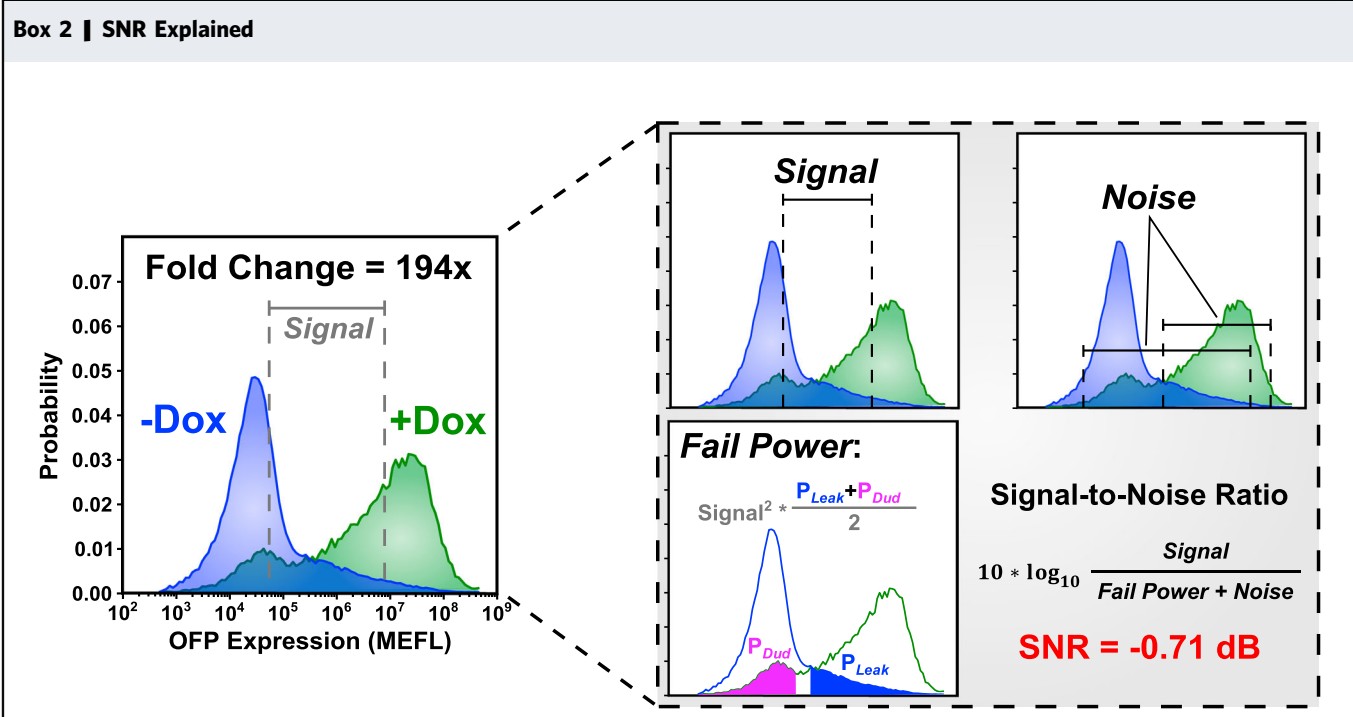

**Definition:** Signal-to-noise ratio, SNR, is the ratio between what we choose to model as signal in the system with what we choose to model as noise in the system. Here we have chosen to model the signal as the difference between the geometric means of the ON- and OFF-states of our digitizer circuits. The noise is broken up into two parts: the average geometric standard deviation between the populations and the "fail power", defined as the sum of the probabilities of a cell being OFF when it should be ON ($P_{Dud}$) or ON when it should be OFF ($P_{Leaky}$).

**Biological relevance:** SNR gives us a metric that describes not only the difference in the average behavior of each circuit state but also considers how the signal is confounded by noise. We define noise in the biological data as the range of expression and the probability of cells falling outside the expected mode of a given population of cells. The higher the difference between average expression of each circuit state and the lower the noise, the higher the SNR value will be. This allows us to precisely describe and compare system performance as we vary the component ratio of each circuit to determine the optimal configuration of our digitizers.

**History:** SNR has been widely used across engineering disciplines and has its roots as a fundamental principle in information theory.

complete induction of all ON state cells. At this ratio, the feedforward circuit topology perform best: the ON-state is clearly distinct from the OFF-state up to 96 h of dox induction. The visual separability of the ON and OFF populations transfected with the feedforward digitizer (Supplementary Fig. S2A) is corroborated by a high AUC value of 0.97 at 96 h, significantly higher than the constant (AUC = 0.76). Examining the SNR of each topology, we see that while the no-shRNA OFP SNR topology peaks at 48 h and dips at 96 h, both the constant and feedforward topologies are able to maintain a constant or increasing OFP SNR between 48 and 96 h post induction (Supplementary Fig. S5A). This indicates that the signal quality of both shRNA topologies is maintained better than that of pTRE-Flp alone, which begins to fail due to leaky Flp translation.

A widely accepted hypothesis in systems biology is that besides having the proper network topology, specific kinetic parameter values are required to achieve desired responses[24]. We hypothesized that there exists some ratio of Flp:shRNA plasmids for each circuit topology that will yield equal performance when optimized. To pinpoint where these ratios lie for each design, we explored a wide parameter space for both shRNA incorporating circuit designs (Fig. 2). A sweep of Flp:shRNA ratios were tested for each circuit topology, and time series data up to 96 h post induction was collected using flow cytometry. As expected, we find that the constant and feedforward topologies produce markedly different response profiles in terms of AUC, SNR, and ΔSNR (i.e., the

difference between the output and input SNR values) when comparing specific ratios, but show similar ability to mitigate noise and produce high signal output when comparing uniquely balanced Flp:shRNA ratios. We find that for the constant topology, a Flp:shRNA ratio of 7:1 yields optimal SNR (3.8 dB) and AUC (0.99) values at 48 h post induction (Fig. 2), with no drop in performance between the 48 and 96 h time points. In addition, this ratio yielded a ΔSNR between at 48 h >0 (5.4 dB), indicating amplification of the input signal (Supplementary Fig. S5B). For the feedforward topology, we note an optimal ratio of 3:1 which yields SNR (3.0 dB) and AUC (0.98) values consistent with a high performing device (Fig. 2). However, we note a decrease in SNR between the 48 and 96 h time points at the 3:1 ratio and instead find the 3:2 Flp:shRNA ratio performs better in terms of maintenance of signal quality over time (Supplementary Fig. S5B). Within the 96 h time frame both the 3:1 and 3:2 feedforward ratios show positive ΔSNR values indicating amplification. Even so, more shRNA may be needed to achieve the optimal balance for the feedforward digitizer when used for longer periods of time. All SNR, AUC, and ΔSNR values are corroborated by associated OFP histogram data, plotted at 96 h for reference (Supplementary Figs. S6 and S7).

Examining the relationship between input and output (transfer curves) for each digitizer topology, we note that these transfer curves may be tuned significantly by varying the ratio of Flp: shRNA. Using the constant topology as a case study, a dox dose-

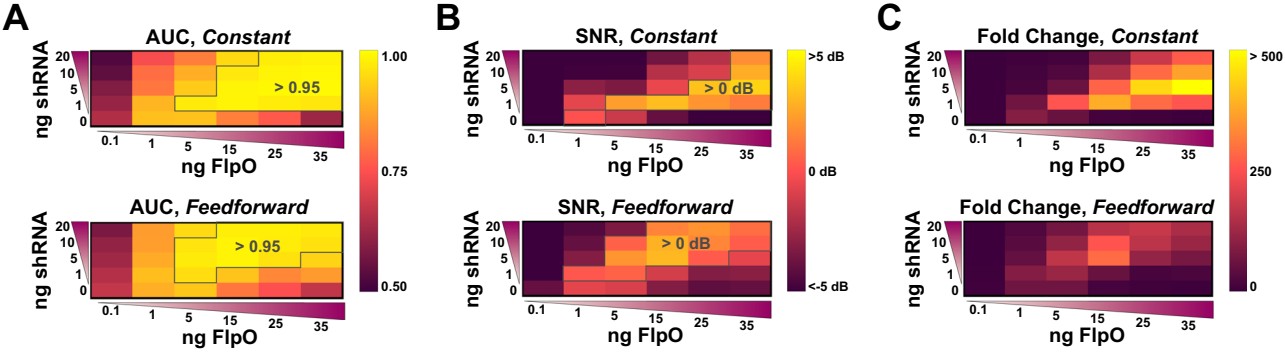

**Fig. 2 Flp:shRNA parameter space highlights divergent performance of each topology.** AUC (**A**), SNR (**B**), and fold change (**C**) heat maps highlight trends among each circuit topology. Outlined sections in the AUC and SNR heat maps represent topologies achieving AUC > 0.95 and SNR > 0 dB, respectively. Data represents the average of three technical replicates ($n = 3$) of circuits transiently transfected into HEK293FT cells 48 h after transfection/induction. Induced cell media contains 225 nM dox.

response curve shows that by titrating different levels of shRNA while holding the amount of Flp plasmid constant we are able to tune the response of each cell population. Increasing levels of shRNA require higher levels of dox induction (i.e., IFP production) to generate any meaningful OFP signal (Supplementary Fig. S8A). Comparing the different digital digitizer topologies, we find that each produces a different transfer curve upon dox induction (Supplementary Fig. S8B). This level of programmability should prove to be a powerful tool for tuning the response of each device given a known level of input, measured by an IFP.

**Biophysical representation of the governing mechanics.** To gain an understanding of how predictable our digitizers are in different contexts and explore the key biophysical reactions governing their behavior, we have built a mixed phenotypic/mechanistic model of each system. To construct our in silico models, we set out to identify the biophysics of each system represented by our metrics. After surveying the literature for previous biophysical models of recombinase behavior we came to several conclusions. None of the previously published models attempt to model shRNA regulation of recombinase behavior in the context of a transient transfection. In addition, a previous model of a serine integrase system that employs a rapid quasi-equilibrium assumption to explore minimal states was able to describe many of the key biophysical bounds of the system, without the need to consider every step in the recombination reaction[25]. We also find that it is not always ideal to make models more complex than they need to be. To model every reaction known to occur in the literature would yield a model too unconstrained to be fit using our collected data. Hence, we chose to follow a similar approach as Pokhilko et al.[25] and build a minimal model to explain our data and then expand on this by evaluating how the addition of complexity affects our model predictions.

There are many possible frameworks we may use to build our model, such as multi-layered methods[26–28] or more classic representations incorporating the law of mass action (LMA) and/or hill equations into sets of stochastic or ordinary differential equations[24,29,30]. We choose to blend these methods by using phenotypic data as an input to the model while keeping to first principles representations of the underlying biophysics of each system using the LMA. By using the input data, we can provide a measure for processes that we have not directly studied and incorporate the variation that exists naturally in the data, which is

crucial for prediction of our signal features represented by the metric-based score each device receives. Our model construction can be broken down into three sections: the testing platform (i.e., transiently transfected HEK293FT cells), cellular reactions, and parameterization.

To capture the effects of our testing platform, we adopt a scheme similar to Davidsohn et al. when building our model where we consider the relative plasmid copy number in transfected cells, rate of plasmid dilution, and initial delay in plasmid transcription due to transient transfection[26]. We created a map of florescence to plasmid copy number using a discrete differential equation whose parameters we fit to constitutively expressing color controls (Supplementary Figs. S9–12). Unlike in previously published work, we find that the time course data for single, constitutively expressed florescent transcriptional units was not fit well by a uniform distribution of initial delay times as would be suggested by cell division being the key driving factor for plasmid uptake (Supplementary Fig. S12). In contrast, we were able to achieve higher precision fits with a Gaussian distribution of initial delay times, which would be consistent with exogenous factors such as cell health playing a role (Supplementary Fig. S12). Also different than Davidsohn et. al., the fit for the plasmid dilution term indicated no observable plasmid dilution over 96 h (Supplementary Fig. S13). While different growth conditions possibly lead to slower cell divisions, the lack of observable plasmid dilution may be due to higher transaction efficiencies where the number of divisions is not sufficient to effect an observable change. All equations and fits for the testing platform components are described in more detail in Supplementary Notes 1 and 2.

To model tyrosine recombinase activity (Flp), we adapted a minimal set of reactions that have been previously shown to model serine recombinases[25]. We combined these with the testing platform equations based on Davidson et al., a law of mass action representation of the shRNA interactions, as well as phenotypic data we collected using proxy measurements for input signal and plasmid copy numbers, fully detailed in Supplementary Note 3. To simplify the resulting set of differential equations, we separated our fast and slow processes and took the limit as the difference between the fast and slow rates approaches infinity. The resulting equation for the change in recombinase over time can then be further simplified by assuming no retroactivity, giving us the final set of equations seen Supplementary Note 3. A complete list of molecular species, reactions, parameters, and ultimately equations we chose to model can be found in Supplementary Tables 1–4. We explored the effects of fitting a more complex model (Supplementary Note 3.4) by adding in the

biochemical states described in Supplementary Figs. S14, S15. However, we found that there was no significant improvement in the fitting the key biochemical parameters using either model, and the non-monotonic behavior given by the added complexity was not observed in the data. Therefore, we chose to move forward with the minimal model.

The model was fit to three data sets displaying a wide range of behavior: maximal expression, leaky expression, and the dose response behavior of all three systems (no-shRNA, constant-shRNA, feedforward-shRNA)(Supplementary Figs. S16–S21). Parameter sensitivity was determined by evaluating the landscape of each fit when starting from different initial conditions. Supplementary Figs. S16 and 17 show the full range of the parameter space explored and display a linear trend in the relationship between the fit minima of each parameter. We use this relationship to further constrain other fits that used the same parameters. The 1D projection of the fitted parameters trajectories shown in Supplementary Fig. S18 indicates all parameters are similarly constrained by the data except $c$, the scaled cleavage and dissociation rate, as the impact of this parameter is not shown in the data until it is varied more than an order of magnitude from its fitted value. More details on parameterization of our model can be found in Supplementary Note 4, and fitted values in Supplementary Table 5. To be thorough, we further explored the sensitivity of several parameters grouped together in our non-dimensionalized system (Supplementary Table 6). Notably, we observed that the parameter grouping $bf$, describing the Flp production rate, and $Kd$, depicting the shRNA imposed threshold on Flp activity, form a linear relationship in their fitted landscape when fit together and while holding all other parameters fixed (Supplementary Fig. S21). This linear relationship is further supported by the grouping of $bf$ into the non-dimensional parameter $eKd$, "effective Kd". As engineers, we may be able to make use of this relationship by turning it into a metric to evaluate tolerance to leaky expression shown in circuit performance. It is also worth noting the eKd can be tied back to the physical balance between concentrations of shRNA and recombinase as depicted in Supplementary Fig. S22. More information on non-dimensional analysis can be found in Supplementary Note 5.

A summary flow chart describing how the model is used to map digitizer inputs to outputs is illustrated in Fig. 3A. By varying the input signal to our model, whether simulated or from experimental data of upstream modules, we can predict the circuit topology and component ratio configuration yielding the desired signal features, such as amplification, as measured by our metrics in terms of FC and SNR. Our results suggest that a higher concentration of Flp is needed to optimally balance small amounts of shRNA in the constant compared to feedforward topologies in order to give equivalent metrics describing behavior. To quantify how well our model captures the exact value of the experimentally observed metrics, we plot experimental values versus predicted values as seen in Fig. 3C and supplementary Fig. S23. We also calculate the Pearson correlation and absolute error of this measurement as seen in Supplementary Table 7. It is worth noting that there is an additional challenge when estimating SNR as calculating this metric requires fitting a mixed Gaussian distribution to define the two modes, which has a level of uncertainty that is common in many non-linear fits. After fitting our model to only a dose response curve for each topology, we are able to predicted the fold change in OFP at 48 h for each system with a high degree of accuracy across a wide range of Flp:shRNA ratios not used in the fitting process (Pearson correlation = 0.90, 0.95, 0.97 for no-shRNA, constant, and feedforward topologies, respectively).

**Creation of cell-cell communication switch.** Synthetic Notch (synNotch) receptors represent a valuable tool for designing cell-to-cell communication systems, offering the ability to tailor their target antigen and induce transcription of a custom output gene in response to antigen binding. However, we find that the syn-Notch module alone displays a weak, graded response between its on and off states when presented with a target antigen (Supplementary Note 8, Supplementary Fig. S24). To demonstrate the value of our rigorous characterization and predictive modeling, we choose to incorporate our digitizers with a synNotch sensor to improve its function as a contact-based cell-to-cell communication switch.

To test our digitizers with the synNotch cell-to-cell communication system, the following representative Flp:shRNA ratios were chosen for both the constant and feedforward modules: well-balanced, over-repressed (too much shRNA regulation) and under-repressed (high leaky Flp expression). The well-balanced state was chosen to be 35 ng Flp, 5 ng shRNA for the constant recombinase module and 15 ng Flp and 20 ng shRNA for the feedforward module. The over-repressed state was chosen to be 1 ng Flp and 5 ng shRNA for the constant recombinase module and 1 ng Flp and 20 ng shRNA for the feedforward module. The under-repressed state was chosen to be 35 ng Flp 1 ng shRNA for the constant recombinase module and 15 ng Flp and 1 ng shRNA for the feedforward module. We also tested a well performing no-shRNA topology of 1 ng Flp 0 ng shRNA. Finally, to analyze the effects of incorporating equal amounts of Flp and shRNA in the constant and feedforward topologies we also tested 1 ng Flp and 5 ng shRNA for the feedforward module to match that of the constant module ratio. Twenty-four hours after transfection of the digitizers into receiver cells stably expressing an αCD19 synNotch sensor releasing a tTA transcription factor, receivers were cocultured with either stable CD19 expressing sender cells or wild type cells at a 1:1 ratio. Flow cytometry was used to look at the CFP, IFP, and OFP expression of these cultures 12 h, 24 h, and 48 h after coculture plating (Fig. 4A). This data was then analyzed to examine the effects of the digitizers on the system behavior in terms of signal quality, amplification, and tolerance to input levels as measured using our metrics seen in Supplementary Table 8.

The population dynamics of the synNotch-driven IFP (Fig. 4B, top) and digitizer OFP (Fig. 4B, middle) reveal that both the constant and feedforward systems increase the population separability compared to the input signal at the balanced ratios. Specifically, we see increases in SNR values between digitizer IFP and OFP (e.g., Δ SNR) of 2.83 dB and 2.85 dB for the constant and feedforward topologies, respectively. These results are corroborated by our model, which predicts increases in OFP SNR using the digitizers (Fig. 4B, bottom). We further note that, as predicted using our model, the over-repressed condition for both topologies fails to induce appreciable receiver OFP production and that the under-repressed conditions produce high levels of basal OFP activation in uninduced receiver cells (Supplementary Fig. S25). Using the model to predict the observed behavior, we see congruence in predicted metrics and actual performance with an absolute error for FC of 1.60, AUC of 1.08 and SNR of 1.60 (Supplementary Table 8) capturing different aspects of the population distributions, as depicted in Fig. 4B and Supplementary Fig. S25.

## Discussion

Here we present a unique characterization and modeling framework that allows for easy programming of genetic circuits. We apply this strategy to tune the properties of three digitizer topologies, which convert an analog input to a digital-like, or Boolean, output. The performance of these digitizers can be tuned by varying the recombinase:shRNA component ratio for (1)

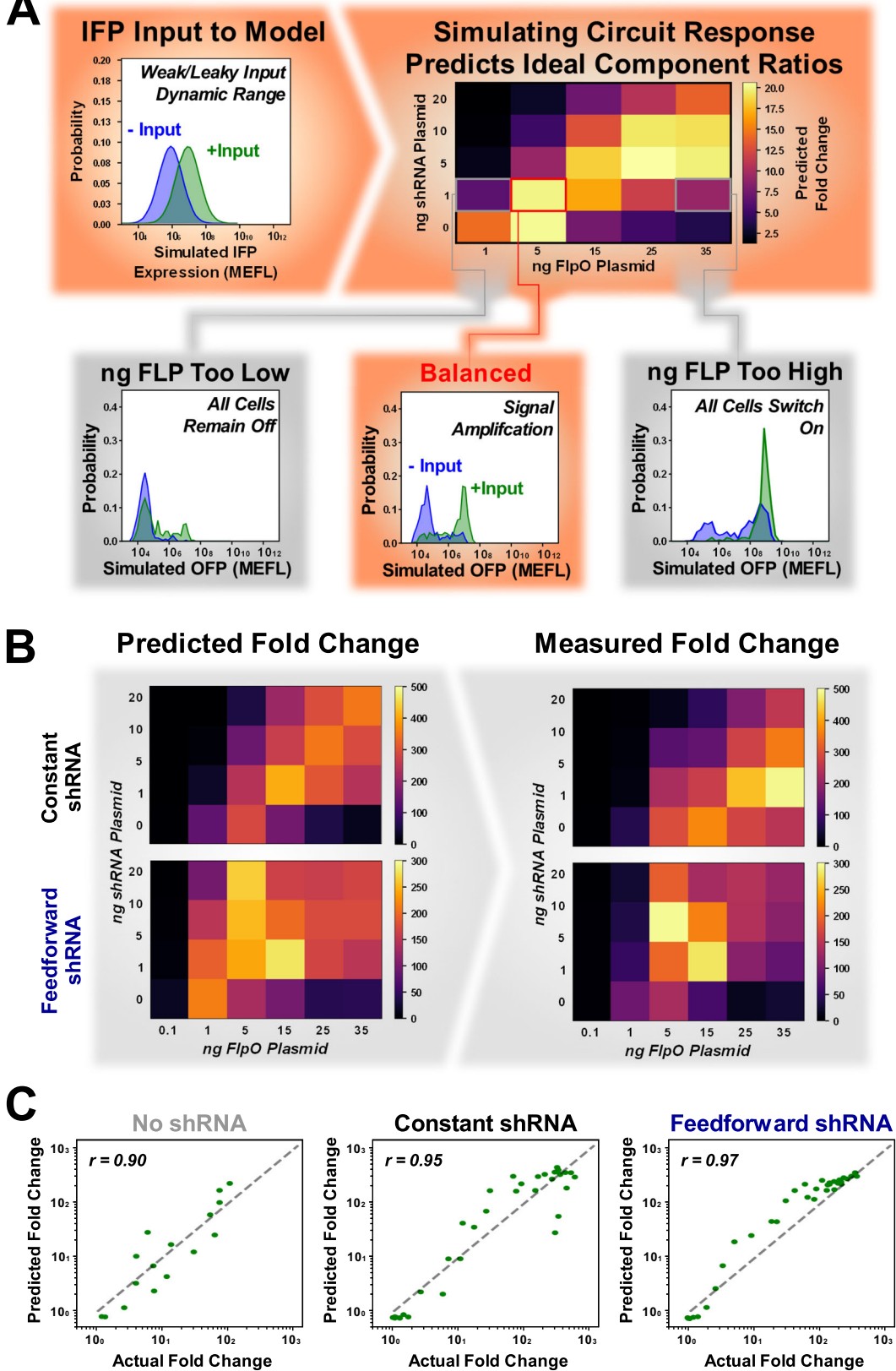

**Fig. 3 Model fit to dose response data predicts time-series data for multiple topologies. A** Flow chart of modeling signal amplification predictions and how these can be mapped back to find optimal Flp:shRNA ratios given a particular input signal. **B** Heatmaps highlighting the trends in digitizer fold change demonstrate similar trends between predicted (left) and experimentally observed (right) values for each topology. **C** Scatter plots of predicted versus actual fold change between plus/minus DOX conditions for each of our three systems at 48 h post transient transfection/induction. Each point represents a different recombinase:shRNA ratio. Number of ratios in each plot is n=16, n=36 and n=36 for the No shRNA, Constant shRNA, and Feedforward shRNA respectively. Pearson correlation coefficient (r) shown at top left of each plot.

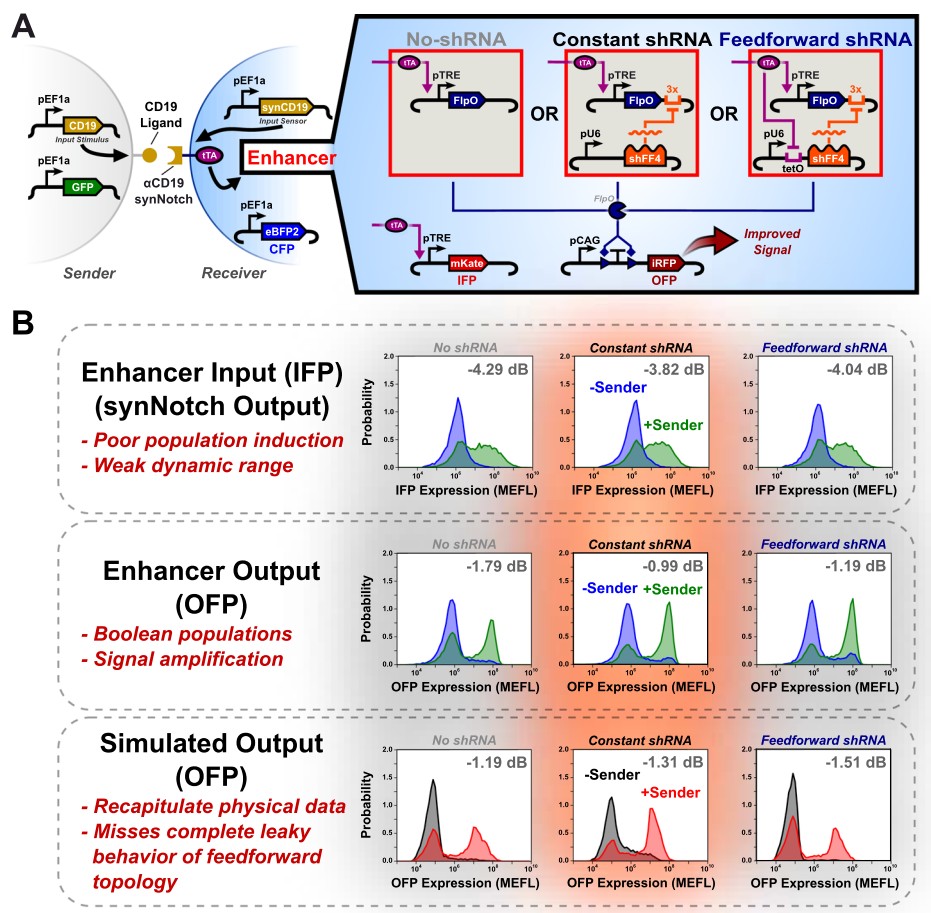

**Fig. 4 Model predicts qualitative enhancement of synNotch signaling by digitizer modules. A** Diagram of the synNotch sensor coupled with each recombinase digitizer topology. **B** IFP (top, synNotch output), experimental OFP (middle, digitizer output), and simulated OFP (bottom) signals demonstrate improved population distinguishability after the signal is transduced by the digitizer. Model results for each digitizer's ability to handle the given input signal from the synNotch sensor qualitatively matches each trend. Experimental data (Digitizer Input and Digitizer Output) represent the top 30% of cells expressing a CFP at 48 h post coculture (72 h post transfection).

maintenance of signal quality, (2) amplification of weak signals, and (3) the ability to mitigate off-target (leaky) component expression. Quantification of signal quality through fold change, AUC, and SNR measurements allows us to clearly define performance standards for each design tested. In addition, our mathematical model, which describes each device's key biophysical properties, provides an in silico map from input to output behavior. We validate the use of these metrics and predictive model by predicting high performing component ratios and failure points for each digitizer. Beyond these insights, we discuss how non-dimentionalization of our model has led to new understanding of device behaviors and describe a metric to evaluate a system's tolerance for leaky expression.

Appropriate metrics are essential for accurately describing system performance. When used effectively they provide specifications for understanding and comparing device behavior. A combination of FC, AUC, and SNR provide an accurate representation of the signal features we aim to program with our digitizers. In addition, our reported ΔFC, ΔAUC, and ΔSNR values measuring the difference between input and output signals provide a measure of how a signal propagates through each device. For example, by comparing the SNR of an input signal

(IFP, controlled by synNotch) with that of an output produced by the digitizer (OFP), we can determine if the signal quality is affected as it is processed by the circuit. The average SNR for the input to the balanced synNotch digitizer systems was −4.05 dB. Comparing this to the average optimized system output of −1.32 dB SNR, we see amplification in our shRNA regulated digitizers (ΔSNR = 2.73 dB).

By fitting our biophysical model to a subset of the experimental data, including the dox dose-response data for each topology at one ratio of recombinase:shRNA, we are able to predict the behavior of each system with a high degree of accuracy for FC and AUC and a modest amount of accuracy when predicting SNR (Supplementary Table 7). We believe the increased variability in predicting SNR comes from an error in fitting the mixed Gaussian model to identify specific populations of cells for the SNR calculation, and not in the model's ability to describe the key signal features of the data. With the added insight we gain from our non-dimentionalization analysis, our model led to both predictive insights for easier circuit module composition as well as a metric, eKd, with which to describe tolerance to leaky expression.

When experimentally optimizing and modeling each topology (no-shRNA, constant-shRNA, and feedforward-shRNA), we note

that each design offers benefits in different ranges of recombinase: shRNA expression as measured by the metrics. At the best ratios tested, both the constant and feedforward digitizers perform nearly identically, maintaining high fold change, AUC, SNR, and ΔSNR values. However, these configurations lie in different regions of the recombinase:shRNA space; when comparing each topology at a particular ratio, the two typically differ greatly in their performance. The constant shRNA topology operates best under conditions of moderate to high recombinase and low to moderate levels of shRNA, while the feedforward topology offers enhanced performance at low levels of recombinase and moderate to high shRNA levels (Fig. 2). Therefore, each system may be useful for different applications in which expression of individual components is limiting.

Many of the methods outlined as part of this work can be applied to a larger set of systems beyond recombinase-based digitizers. Incorporating phenotypic data into minimal model generation is a relatively new strategy that we believe will have lasting impacts on how we interface mechanistic models with complex biological systems. Finding relevant and descriptive metrics for circuit performance will also help the field as a whole become more standardized in its approach to designing biological systems. While we apply the SNR and AUC metrics to single-cell flow cytometry data in this work, we anticipate that these metrics will be useful for bulk measurements such as luminescence data from a plate reader or fluorescence microscopy images.

In summary, this work demonstrates the advantage of using metrics and modeling to evaluate and predict the performance of digitizer circuits. We are the first to describe quantitative amplification using digitizers, and show that the performance of these digitizers is predictable in silico. This predictability allows for their incorporation in the design of complex systems to improve their signal processing. The value of the characterization workflow provided here is highlighted by the composition of each digitizer topology with a synNotch sensor to create an improved cell-to-cell communication switch. These switches demonstrate predictable amplification of a weak synNotch input signal and improve its distinguishability without the need for iterative optimization. We posit that other gene circuits consisting of modular components will benefit from the use of standard metrics to evaluate their performance, including SNR, AUC, and FC. We hypothesize that this will lead to more reliable use of component parts in new contexts.

## Methods

The methods were performed in accordance with relevant guidelines and regulations provided by the both the Boston University and Massachusetts Institute of Technology Environmental Health and Safety Departments.

**Cell maintenance**. All experiments were performed using the HEK293FT cell line purchased from ATCC. Cells were maintained in DMEM medium (Corning) supplemented with 5% heat-inactivated fetal bovine serum (FBS) (Gibco), 50 IU /mL penicillin, 50 μg/mL streptomycin (Corning), 2 mM L-glutamine (Corning), and 1 mM sodium pyruvate (Lonza) (5PSGN). For all experiments using the Tet ON or OFF systems, cells maintained in 5PSGN were transitioned to a similar media formulation (5PSGN-T) using Tet-Approved FBS (Clontech).

**Cell plating, transfection and induction**. One day prior to transfection, HEK293FT cells were trypsinized using 0.05% trypsin, 0.53 mM EDTA (Corning) for 3 minutes at 37 °C and neutralized by adding 5PSGN in a 3:1 media:trypsin ratio. Cells were pelleted, resuspended into a single cell solution using fresh 5PSGN-T, and aliquots were transferred to 48-well plates (250 μL/well, 250k cells /mL). After transfer, plates were shaken in a jerking motion to evenly disburse the cells.

Each well of a 48 well plate was transfected with 250 ng total DNA using polyethylenimine (PEI) and plasmids of similar size to help produced desired ratio of components; transfection information, including plasmid amounts per tranfected sample, can be found in our repository on SynBioHub (see Data

Availability for link, Supplementary Note 7 for more information). PEI stocks were made using linear PEI (Polysciences 23966, MW = 25k) dissolved with the assistance of concentrated hydrochloric acid and sodium hydroxide to a concentration of 0.323 g /L in deionized water and then filter sterilized (0.22 μm). Stocks were stored at −80 °C until use, and were stored at 4 °C after thawing. All individual transfection mixes (1 μg total DNA, 50 ng/μL) were brought to 50 μL total volume using a 0.15 M sodium chloride solution. Separately, 8 μL PEI stock solution was diluted into 42 μL 0.15 M sodium chloride solution. DNA and PEI mixtures were vortexed to mix, combined (100 μL total volume), vortexed again and spun down. In total, 25 μL of each transfection mix was pipetted onto three wells of a 48 well plate ($n = 3$).

Doxycycline (Dox) (Millipore Sigma) was added immediately after transfection in all experiments. Dox was dissolved in ethanol to make a 1000x stock solution at 100 μg/mL. For all time series experiments, the stock solution was diluted 7:125 into 5PSGN-T, and 5 μL of the diluted stock was added to each ON state well for a final working concentration of 100 ng/mL. For all Dox dose response experiments, the stock solution was diluted to 56x each concentration used in the titration. In total, 5 μL of each dilution was then added to triplicate wells.

Toxicity and resource effects were also explored by looking for a change in CFP production as a results of higher dox addition as depicted is Supplementary Fig. S26. Impact of these effects is discussed further in Supplementary Note 8.

**Coculture experiments**. For coculture experiment cells were transiently transfected the day before coculture in adherent cell culture. Using Versene to release cells from the plate without cleving off extra cellular domains, cell suspensions were made from wild type, sender and receive populations and 90 cell/ml. Receiver cells where then coCultured with either sender or wild type cells in a 24 well plate at a 1:1 ratio (250 μl of each cell suspension was used) Cell were then allowed to grow for an additional 48 h and florescence values were collected using flow cytometry.

**FACs configuration**. All data for experiments characterizing digital digitizer performance were collected using an Attune NxT flow cytometer with an attached Autosampler (Life Technologies). The Attune was equipped with violet (405 nm), blue (488 nm), yellow (561 nm) and red (605 nm) excitation lasers and the default filter configuration for each (http://tools.thermofisher.com/content/sfs/manuals/ 100024236_AttuneNxT_SW_UG.pdf). Specifically, the Attune is equipped to measure eGFP (488 nm laser, 530/30 emission filter, 555 nm dichroic longpass mirror), mtagBFP2 (405 nm laser, 450/50 nm emission filter, 495 nm longpass dichroic mirror), mRuby2 (561 nm laser, 585/16 emission filter, 600 nm dichroic longpass mirror), iRFP720 (605 nm laser, 720/30 nm emission filter, 740 nm dichroic longpass mirror), and LSSmOrange (405 nm laser, 603/48 nm emission filter, 650 nm dichroic longpass mirror). Cells were trypsinized (0.05% trypsin, 0.53 mM EDTA, Corning) half an hour before data collection and resuspended in 5PSGN medium to generate a single-cell suspension for flow cytometry. After resuspension, all samples were transferred from 48 well plates to 96 well plates to run using the Attune Autosampler. Time series data were collected individually at each time point, and all other data were collected 48 h post induction.

Experimental data for the synNotch sensor module, both transient and integrated, as well as the composition of the sensor and digitizer modules were collected using the BD LSRFortessa flow cytometer. This instrument was equipped to measure eGFP and eYFP (488 nm laser, 530/30 emission filter), mtagBFP2 (405 nm laser, 450/50 nm emission filter), mKate (561 nm laser, 610/20 emission filter), iRFP720 (640 nm laser, 780/60 emission filter). Cells were trypsinized, resuspended and run on the Fortessa in a similar manner as described using the Attune.

To generate a compensation model for all experiments, a standard set of controls was run with each experiment: a single positive control for each fluorescent protein collected (EGFP, mtagBFP2, and mRuby2), a multicolor control consisting of each individual single color control transfected at an equal plasmid weight, a blank control of cells transfected with a null plasmid, and a set of 3.0 μm SPHEROTM Rainbow Calibration Particles (Lot AE01, Spherotech Inc.)

**TASBE analysis**. A flow chart of the TASBE process including the gating strategy for the FACs data is provided in Supplementary Fig. S27. The use of fluorescent proteins as a proxy for component performance within our devices requires that the production of these proteins be quantitatively comparable. To ensure this, we make use of the TASBE Flow Analytics package available through the open source software development platform Github (https://github.com/TASBE/TASBEFlowAnalytics). For each experiment, the standard set of controls detailed in Data Collection was used to convert all raw fluorescence data to a standard Molecules of Equivalent Fluorescein (MEFL) unit. All control data is collected at 48 h post transfection, and MEFL calibration is performed as described by Beal et. al.[31]. Briefly, autofluorescence is first corrected for by subtracting a normal distribution fit of flow cytometry data collected from HEK293FT cells transfected with a null plasmid (BW363) from all other raw data. A 2D Gaussian mixture model is then computed for this same blank data set, and the component containing live, single cells is used to gate all other samples. After gating and correcting for autofluorescence, linear compensation is performed for each fluorescence channel using the single positive control data for each fluorescent protein. Next, a linear conversion is computed for all colors with respect to EGFP, in

which mtagBFP2 and mRuby2 arbitrary units are converted to EGFP arbitrary units using data from the multicolor control. These EGFP arbitrary units are then converted to MEFL using flow cytometry data collected from the beads standard.

**Statistics and reproducibility**. As indicated throughout the article, three technical replicates were collected for all flow cytometry unless otherwise indicated. Geometric mean and standard error of the mean, S.E.M. where used is all line plots. Representative technical replicates were used as indicated in all histogram plots. Pearson correlation coefficients were calculated using the Python 3 Scipy Package 1.5. Calculation of the FC, AUC and SNR metrics are described in our results section Metrics for Functional Characterization of Digitizer Modules. All data sets used in figure generation as well as analysis presented in this work are available for download in our SynBioHub repository (https://synbiohub.programmingbiology.org/public/DigitizingCommunication/DigitizingCommunication_collection/1).

All simulations were done using custom code via Python 3. All images were assembled using Inkscape. All Flow Cytometry Data was processed with TASBE Flow Analytics package available through the open source software development platform Github (https://github.com/TASBE/TASBEFlowAnalytics). For each experiment, the standard set of controls detailed in Data Collection was used to convert all raw fluorescence data to a standard Molecules of Equivalent Fluorescein (MEFL) unit.

**Reporting summary**. Further information on research design is available in the Nature Research Reporting Summary linked to this article.

## Data availability
Data for all figures and results presented here, as well as plasmid maps for all plasmids used in this study, are available for download from the SynBioHub repository at: https://synbiohub.programmingbiology.org/public/DigitizingCommunication/DigitizingCommunication_collection/1[32].

## Code availability
Modeling code is available on Zendo at https://doi.org/10.5281/zenodo.4758500.

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

## Acknowledgements
K.K. and R.W. acknowledge funding from NSF Expeditions in Computing Award (1521925) part of the Living Computing Project (https://www.programmingbiology.org/) J.H.L. acknowledges funding from the NHLBI division of the NIH (F31HL149334) and the NSF Expeditions in Computing Award (1522074), part of the Living Computing Project (https://www.programmingbiology.org/). B.H.W. acknowledges funding from the NSF Expeditions in Computing (1522074) and NSF Career (162457) Awards. W.W.W. acknowledges funding from the NIH Director's New Innovator Award (1DP2CA186574), NSF Expeditions in Computing (1522074), NSF CAREER (162457), NSF BBSRC (1614642), and NSF EAGER (1645169). J.B. acknowledges funding from the NSF Expeditions in Computing (1522074) grant. L.W. and C.M. acknowledge funding from NSF Award Number 1748200. Any opinions, findings, and conclusions or recommendations expressed in this material are those of the author(s) and do not necessarily reflect the views of our funding agencies.

## Author contributions
K.K. Conceptualized ideas, designed experiments, collected data, developed methodology for model and paper writing. J.L. Conceptualized ideas, designed experiments, collected majority of experimental data and paper writing. B.W. Conceptualized ideas and designed experiments. J.W. Developed methodology for model and paper writing. M.C. Synbiohub entry data curation and paper writing. L.W. Developed methodology for model. C.M. Supervized synbioHub efforts and model development. J.B. Contributed to metric design and supervised model development and experimental design. W.W. Supervised model development and experimental design. R.W. Supervised model development and experimental design.

## Competing interests
W.W.W. has consulted for and own shares in Senti Biosciences. The other authors declare no competing interests.
