## [Peer Review File · Communications Biology]

Reviewers' Comments:

Reviewer #1:

Remarks to the Author:

I read this manuscript with great interest. It provides a clean workflow to conduct and analyze experiments using recombinases and pTRE/dox repression system in eukaryotic cells. However, there are several issues in the manuscript that need to be addressed before it could be published.

* Between lines 187-195, the authors stated that 5ng Flp to 50ng shRNA ratio provides the best separation between ON and OFF state. On line 410, the authors stated that 250ng total DNA was transfected into each well on a 48-well plate. I want to know the exact amounts of plasmids were used for transfection in "No shRNA Regulation" group - how much pCAG-stop-GFP (also pCAG-mRby), pEF1a-rtTA and pTRE-BFP, any pU6 driven scramble sequence.

* Plasmid U6tetO-shRNA was used in "Feedforward shRNA" design, and it did increase FC comparing with "Constant shRNA" design. Is it possible to experimentally show that after adding 225 nM dox the shRNA expression driven by U6tetO was reduced, which later result in higher Flp expression and followed by greater GFP signal?

* In addition to FC, the authors propose to use AUC and SNR as new metrics to report the system readouts. When optimizing Flp:shRNA ratio (Figure 2), even if there is no significant change of FC, it still should be documented/presented. I think, how generalizable of these two metrics (AUC/SNR) for synthetic biology, not limited to this study (i.e. for studies without access to single-cell resolution data), should be discussed.

* Line 178: the reason for only analyzing the top 30% CFP expressing cells is to compare cells transfected with similar amounts of DNA. However, in Figure S3, OFP SNR values, but not CFP value were shown. I am also confused about the trends of 5ng Flp plasmid (top left) top 20% transfection with 15-20ng shRNA.

* With Figure 3 showing the accuracy of the model, Table 1 (related Figure 4B, three different Flp:shRNA ratio) is confusing and I would move it to the supplementary material.

* It takes time for recombinases to have effects, and IFP signal turns on earlier than OFP signal. Comparing IFP and OFP at the same time point (as in Figure 4B) might not be fair. Maybe the authors could discuss it in the main text.

Reviewer #2:

Remarks to the Author:

In their paper Kiwimagi et al. report on the use of parallel experimental and theoretical characterisation for optimised design of digitizer circuits.

The main aim is focused on ensuring the ability to tune shRNA-based digitizers in their performance according to noise level, leakiness and dynamic range.

The paper content is valuable and original, and I thus suggest publication. One aspect that has not been evaluated by the authors is how combining their strategy with different construct design alternatives could further improve the system. For example, would the use of degradation tags on the Flp protein help decreasing leakiness in the system? could this improve SNR?

Other minor comments:

I find that when citing figures in the main text and in the supplementary file, authors do not keep consistency and use Fig., Fig, figure, figure (). I would suggest editing this. For example: supplementary file line 120, 146, 171 and throughout.

In the main text authors refer to Fig S2A and B but I can't really find it. Apologies if it is my mistake but I have the impression they refer to Fig S1 instead.

I would suggest reporting in Fig S1, S7 and S8 details of the experimental conditions.

In Fig 2, if I understood correctly, I believe the side yellow bar should have a purple to yellow color shading which is missing.

In supplementary file line 404 a reference is missing.

Reviewer #3:

Remarks to the Author:

Kiwimagi, Letendre and co-workers implemented recombinase-based digitizer circuits in a Human embryonic kidney cell line. They performed a thorough characterization of 3 different topologies of the circuit. They implemented a new set of metrics that quantitatively describe circuit behavior and that can be applied to other biological circuits. They developed a mechanistic model that predicted circuit performance and they applied the workflow to generate a synthetic cell-cell communication device.

The presented work is novel and it is of relevance to the field. Specially to introduce new metrics that allow better description and evaluation of biological circuit design and performance. One of the major claims of the work is to be able to quantitatively define digitizer performance and predict responses, they achieve so by using the new set of metrics and by implementing a mechanistic model. They improved the leakiness of the initial design and showed how signal is amplified in the cell signaling case.

Major comments

I find the mechanistic modeling part of the manuscript to need drastic improvements. The results section limits to describe the background and only refers to the supplement, then they go straight to the obtained results without providing an explanation on what are the key variables of the model, how they performed the parameterization, the obtained values for those key parameters and the sensitivity of this parameters for the obtained results of the simulations. It may be desirable to also showcase how this model can be employed to describe other published circuit designs, and what are the measurements of variables needed to use the model in other contexts.

Some of the presented data do not show error or standard deviation, significance test, n, etc.

The molecular weight of each plasmid, since you are using ng as a unit of copies of circuits. Are plasmid sizes equivalent? How does plasmid size affect your calculations?

How do you define the on/off threshold for the AUC calculations? Please provide a definition, this is key to reproduce the calculations

All the parameters used in the model need to be clearly presented for example: shRNA silencing of FLP expression.

Further explain why authors use top 30% transfected cells, this may be obvious but need clearer explanation

Minor:

The whole manuscript can be improved and be clearer for better understandability, I consider this to be of major importance in this study that may be read by people from different disciplines (engineers and biologists) thus should be clearer in their writing. I think that adjusting to space restrictions of the journal, some parts of the supplementary text may be added to the main text for better description of the system. For example, the description and background of the synNotch system, the description of the model in the main text is limited to : "As detailed in the supplement...."

Something that may be obvious for an engineer but is not for a biologist is that as far as I understand, the goal of a digitizer is to convert an analog signal to a digital one, which is the

analog signal and the digital output in this case? Why is this important? explain to biologists

Lane 213: What do you mean by signal handling capacity?

All acronyms should be clearly defined in the main text or methods (e.g. lane 82 FSF-GFP is not defined)

Dox toxicity issue. To show an experiment with the CFP cell line with increased dox concentration but without the IFP plasmid, this could clarify if it is the dox or the resource competition for both proteins as has been shown in many works in bacteria

Some of the figures need bigger text legends

Some data mentioned in the text is in a different figure than referenced, make sure all data is correctly presented

The whole text need to be revised, as it has many typos and errors here are some that I can point:

Sfigure 2 - Dose response histogram

S figure 1- time series data

Line 78-80: codon optimized flippase is referred as Flp, lane 133 and many other parts of text and figures is referred as FlpO, this is confusing, clarify if it is the same or different proteins.

Figure 2: the color scale used for heatmaps is only one color

Sfigures 12-27 are not mentioned in the main text

Lane 175: palsmid

Lane 340: dimentioanlization

Reviewer 1: Response to Comments

Reviewer Comment	Response
I read this manuscript with great interest. It provides a clean workflow to conduct and analyze experiments using recombinases and pTRE/dox repression system in eukaryotic cells. However, there are several issues in the manuscript that need to be addressed before it could be published.	
* Between lines 187-195, the authors stated that 5ng Flp to 50ng shRNA ratio provides the best separation between ON and OFF state. On line 410, the authors stated that 250ng total DNA was transfected into each well on a 48-well plate. I want to know the exact amounts of plasmids were used for transfection in "No shRNA Regulation" group - how much pCAG-stop-GFP (also pCAG-mRby), pEF1a-rtTA and pTRE-BFP, any pU6 driven scramble sequence.	Thank you for the note - we have added a section in the Methods about our repository in SynBioHub, where all of this data can be found for each experiment in the form of Excel spreadsheets for each experiment: Methods>>Cell Plating, Transfection and Induction>>2nd paragraph: ..."transfection information, including plasmid amounts per transfected sample, can be found in our repository on SynBioHub (see Data Availability for link)." We are hesitant to include all of this information in the main text as we feel it will draw the reader's attention away from the main points of discussion - the utility of the digitizers, and the specific ratios of Flp:shRNA plasmid that are used to achieve our desired responses
Plasmid U6tetO-shRNA was used in "Feedforward shRNA" design, and it did increase FC comparing with "Constant shRNA" design. Is it possible to experimentally show that after adding 225 nM dox the shRNA expression driven by U6tetO was reduced, which later result in higher Flp expression and followed by greater GFP signal?	Thank you for addressing this point - supplemental figure 8 highlights the notably different response profiles of the Constant shRNA digitizer and Feedforward shRNA digitizer topologies, and we feel this appropriately shows that shRNA is indeed being repressed by rtTA binding (due to dox induction). The TET operator site in between the U6 promoter and shRNA transcript is the only variable that changes between the two digitizers, and we note that the Feedforward digitizer yields a transfer curve that is almost linear in its response to doxycycline: as more IFP is measured (a proxy for rtTA binding/pTRE activation), more OFP is produced by the digitizer. For the Constant digitizer topology, a very high level of IFP is needed to begin observing an OFP signal, and we hypothesize this is due solely to the constant expression of shRNA.
In addition to FC, the authors propose to use AUC and SNR as new metrics to report the system readouts. When optimizing Flp:shRNA ratio (Figure 2), even if there is no significant change of FC, it still should be documented/presented. I think, how generalizable of	Thank you for the suggestion on including fold change to compare with AUC and SNR. We have included fold change as panel C in the figure, and now explicitly compare the distribution of 'acceptable' circuits based on fold change, AUC and SNR. With regard to the generalizability of these metrics, we hope that other researchers will employ them as they are easily

these two metrics (AUC/SNR) for synthetic biology, not limited to this study (i.e. for studies without access to single-cell resolution data), should be discussed.	adaptable to other systems. We discuss this further in an updated version of our Discussion section (Discussion>>5th paragraph)
Line 178: the reason for only analyzing the top 30% CFP expressing cells is to compare cells transfected with similar amounts of DNA. However, in Figure S3, OFP SNR values, but not CFP value were shown. I am also confused about the trends of 5ng FIp plasmid (top left) top 20% transfection with 15-20ng shRNA.	Thank you for the note - we realized there was an error in the code pulling the incorrect SNR values for the top 20% of transfected cells for the 5 ng FIp plasmid plot. This has been corrected, and the data points for the top 20% transfected cells qualitatively matches more closely the trends seen in all other transfection bins. Additionally, we have adjusted the legend to be more clear about what the reader is looking at (i.e. transfection bins based on CFP expression) and made the text larger
With Figure 3 showing the accuracy of the model, Table 1 (related Figure 4B, three different FIp:shRNA ratio) is confusing and I would move it to the supplementary material.	This table has been moved to the supplemental. (new Supplemental Fig. S7)
It takes time for recombinases to have effects, and IFP signal turns on earlier than OFP signal. Comparing IFP and OFP at the same time point (as in Figure 4B) might not be fair. Maybe the authors could discuss it in the main text.	Thank you for your comment. We wanted to compare the performance of a circuit with and without a digitizer under the same conditions including incubation time. With this in mind we did not want to give one circuit more of an advantage even though there is an increase in the number of reactions. It is true there will be a delay between IFP and OFP in reaching some kind of steady state. We indirectly looked at this when trying to determine the leakiness of these system overtime. We observed many conditions at 6,12 48 and 96hours, a subset are plotted in Supplementary Figure S2. The majority of change happens between 12 and 48hours. The little change between 48 and 96hours does suggest the both systems have reached a "quasi-steady state" in this transient experiment. If we wanted to study the delay due to the added states we would need to run a more fine time course experiment looking at what happens between 12 and 48 hours which we believe is outside the scope of this study.

Reviewer 2: Response to Comments

Reviewer Comment	Author Response
In their paper Kiwimagi et al. report on the use of parallel experimental and theoretical characterisation for optimised design of digitizer circuits. The main aim is focused on ensuring the ability to tune shRNA-based digitizers in their performance according to noise level, leakiness and dynamic range. The paper content is valuable and original, and I thus suggest publication.	
One aspect that has not been evaluated by the authors is how combining their strategy with different construct design alternatives could further improve the system. For example, would the use of degradation tags on the Flp protein help decreasing leakiness in the system? could this improve SNR?	Thank you for the suggestion. While reducing the level of Flp protein in general may give a similar effect to what we have done in our system using shRNA specifically, we believe this is outside the scope of this work - our goal was not to account for every possible circuit design, but rather create a framework for rigorously characterizing circuits so that they may be used more interchangeably. We hope that readers will be encouraged to study these systems further for their own use, incorporating unique designs like the one you mention here.
Other minor comments: I find that when citing figures in the main text and in the supplementary file, authors do not keep consistency and use Fig., Fig, figure, figure (). I would suggest editing this. For example: supplementary file line 120, 146, 171 and throughout.	Thank you for bringing this to our attention. All references to figures should now read Fig. N
In the main text authors refer to Fig S2A and B but I can't really find it. Apologies if it is my mistake but I have the impression they refer to Fig S1 instead.	Thank you for pointing this out - the dox dose response and time series (formerly figures S2 and S1, respectively) were in the wrong order - these have been swapped to reflect the correct order.
I would suggest reporting in Fig S1, S7 and S8 details of the experimental conditions.	Thank you for this suggestion - experimental details have been added to all supplemental figures if not previously included
In Fig 2, if I understood correctly, I believe the side yellow bar should have a purple to yellow color shading which is missing.	Thank you for bringing this to our attention - yes, somehow the file we uploaded did not display the correct shading for these color bars and the final figure also had some missing information in Panel A. These have both been adjusted by changing the file type of the figure embedded in the pdf
In supplementary file line 404 a reference is missing.	This has been added - thank you.

Reviewer 3: Response to Comments

Reviewer Comment	Author Response
Kiwimagi, Letendre and co-workers implemented recombinase-based digitizer circuits in a Human embryonic kidney cell line. They performed a thorough characterization of 3 different topologies of the circuit. They implemented a new set of metrics that quantitatively describe circuit behavior and that can be applied to other biological circuits. They developed a mechanistic model that predicted circuit performance and they applied the workflow to generate a synthetic cell-cell communication device. The presented work is novel and it is of relevance to the field. Specially to introduce new metrics that allow better description and evaluation of biological circuit design and performance. One of the major claims of the work is to be able to quantitatively define digitizer performance and predict responses, they achieve so by using the new set of metrics and by implementing a mechanistic model. They improved the leakiness of the initial design and showed how signal is amplified in the cell signaling case.	
Major comments: I find the mechanistic modeling part of the manuscript to need drastic improvements. The results section limits to describe the background and only refers to the supplement, then they go straight to the obtained results without providing an explanation on what are the key variables of the model, how they performed the parameterization, the obtained values for those key parameters and the sensitivity of this parameters for the obtained results of the simulations. It may be desirable to also showcase how this model can be employed to describe other published circuit designs, and what are the measurements of variables needed to use the model in other contexts.	Thank you for bringing this up. We have now summarized the 5 supplementary notes, written to explain the construction of our model, in the main text, including explanation of supplementary figures and tables containing all our parameters, the sensitivity analysis of the fits as well and non-dimensionalization analysis done as part of the parametrization of the model. The text can essentially be broken down into three categories: Modeling the testing platform, molding the cellular reactions and parameterizing the model summarized in three paragraphs found between line 250 and 306. We also took time to add to the main text highlights of the parameter sensitivity in fitting that led to key insights regarding the relationship between the parameters and how that information is useful to the biological engineer. To respond to the request to showcase how this model could be applied to describe other published systems, we added a description to the discussion to address the kind of impact we believe an effort to incorporate minimal state modeling and phenotypic data will have on the field. Although outside the scope of this work our excitement for how in the future these methods could go beyond recombinase-based digitizers.

Some of the presented data do not show error or standard deviation, significance test, n, etc.	We have added error bars representing the SEM to the following figures: 2B, 4, 5, 13A
The molecular weight of each plasmid, since you are using ng as a unit of copies of circuits. Are plasmid sizes equivalent? How does plasmid size affect your calculations?	Thank you for bringing this comment. Yes, plasmid size does effect the transfection efficiency of each plasmid. We have been careful to use plasmids of equivalent size in our experiments to reduce this confounding factor in our analysis. We have also measured how well our plasmids of equivalent size are correlated in there transfection, Supplemental Figure S12 and found congruent results to what has been previously published as cited in our Supplementary note 1.1. We have added a few words to the methods section to help clarify this effect: Methods>>Cell Plating, Transfection, and Induction>>paragraph 2: "... and plasmids of similar size to help produced desired ratio of components..."
How do you define the on/off threshold for the AUC calculations? Please provide a definition, this is key to reproduce the calculations	Thank you for bringing this to our attention - we have been more explicit about the thresholding when determining AUC, but in short there is no defined threshold. We vary where the on/off threshold is drawn to generate the ROC curve and the area under this curve (AUC) is calculated to determine how distinguishable (i.e. separable) induced and uninduced cell populations expressing the digitizer circuits are from one another. A completely distinguishable circuit, where at some arbitrary threshold value we capture all true positive events while excluding all false positive events (i.e. all cells induced are ON, all cells uninduced are OFF), would look like a perfectly separable system whose AUC = 1.0. To help make this more apparent to our reader we have brough our AUC description BOX from the supplemental to the main text (line 145).
All the parameters used in the model need to be clearly presented for example: shRNA silencing of FLP expression.	Thank you for stressing this point. We have added direct references to our supplementary tables that describe each parameters used in the model. Sup tables ?????
Further explain why authors use top 30% transfected cells, this may be obvious but need clearer explanation	Thank you for the note on this - we have updated the wording in the first section of Results, but in short we focus on the top 30% of transfected cells because 1) trends among different transfection bins are qualitatively similar, so looking at different transfection bins would not significantly impact our findings and 2)

	we focus on this particular transfection bin to analyze cells that are transfected with similar amounts of plasmid for a quantitative comparison of digitizer systems. Updated text can be found in Results>>Design of Digitizer>>2nd paragraph
Minor: The whole manuscript can be improved and be clearer for better understandability, I consider this to be of major importance in this study that may be read by people from different disciplines (engineers and biologists) thus should be clearer in their writing. I think that adjusting to space restrictions of the journal, some parts of the supplementary text may be added to the main text for better description of the system. For example, the description and background of the synNotch system, the description of the model in the main text is limited to :” As detailed in the supplement.....”	Thank you again for the feedback. As to the clarity of the manuscript we have made several changes to the wording, especially in description of the model. We have also added more references to our detailed description of the synNotch system. This did push us over the word limit but we think it makes the paper much better. We have also made edits throughout the manuscript for clarity with a good amount of attention given to the introduction to help provided more background given the wide target audience of the paper. (We describe each change in the line by line edits description below.)
Something that may be obvious for an engineer but is not for a biologist is that as far as I understand, the goal of a digitizer is to convert an analog signal to a digital one, which is the analog signal and the digital output in this case? Why is this important? explain to biologists	Thank you for your feedback. To answer your questions endogenous biological systems contain many types genetic circuit generated transcriptional outputs including circuits with a graded responses, analog signal, and circuit with a strict set for on and off behavior, digital signal. Both signal types are important for biological systems to function and the conversion from analog to digital is key for biological systems to make distinct decisions given the often graded input they receive from their soundings. When we can strategically engineering these digital responses into our synthetically engineered circuits we are given the ability to program discrete decisions based on a graded input. In response to your questions we have also added a deeper explanation in the 2nd paragraph of the introduction to address this point, and outline which signals are analog and digital in the 1st paragraph of the Design of Digitizer section in Results
Lane 213: What do you mean by signal handling capacity?	Thank you for pointing this out - we have used more clear phrasing here. In short, it refers to how a signal will transduce through the device, but we agree that the original phrasing of 'signal handling capacity' was unclear - this has been changed to "...ability to mitigate noise and produce high output signal..." in Results>>Metrics for Functional...>>6th paragraph

All acronyms should be clearly defined in the main text or methods (e.g. lane 82 FSF-GFP is not defined)	Thank you for pointing this out - we have tried to be more explicit as to what all of the acronyms in the paper refer to. (ex. Ordinary Differential Equation, ODE)
Dox toxicity issue. To show an experiment with the CFP cell line with increased dox concentration but without the IFP plasmid, this could clarify if it is the dox or the resource competition for both proteins as has been shown in many works in bacteria	Thank you for the suggestion here - while the experiment you proposed would get at the question for the specific case of a constitutive promoter, we feel it is likely that both toxicity and resource competition affect our system (outlined in Supplemental Section 1.6). We do not believe this effect to have a significant impact on our results due both the small fold change difference of 2 fold reported in supplementary note 1.6 as well as the clear trends and predictable behavior of the system throughout the manuscript. To be thorough in supplementary note 1.6, we did address the question: how could toxicity or resource competition affect the output of our digitizer topologies. We hypothesize how this might affect the discrepancies in our modelling vs experimental results when composing digitizers together with synNotch sensors. We note that in the shoulder of some of the disruption there is some discrepancy between the predicted and actual results of composing these elements together, and we hypothesize this is likely due to toxicity rather than resource competition as the absence of doxycycline is the significantly different variable when using synNotch as an input rather than dox-mediated rtTA.
Some of the figures need bigger text legends	This has been updated as noted in our line by line edits below
Some data mentioned in the text is in a different figure than referenced, make sure all data is correctly presented	This has been fixed - thank you for bringing this to our attention.
The whole text need to be revised, as it has many typos and errors here are some that I can point:	Applicable to all below (shown as “): Thank you for addressing the spelling and grammar concerns - we have thoroughly reviewed and edited the main and supplemental text to address these concerns
Sfigure 2 - Dose response histogram	"
S figure 1- time series data	"
Line 78-80: codon optimized flippase is referred as Flp, lane 133 and many other parts of text and figures is referred as	This has been adjusted to read 'Flp'

FlpO, this is confusing, clarify if it is the same or different proteins.	
Figure 2: the color scale used for heatmaps is only one color	Same comment as above, already addressed
Sfigures 12-27 are not mentioned in the main text	Thank you - these are now referenced in our updated and more comprehensive modelling section in the main text
Lane 175: palsmid	"
Lane 340: dimentioanlization	"

All Edits to the Manuscript

Section	Paragraph	Edits
Introduction	line 1-17 (1st paragraph in introduction)	In response to the reviewers comments we have reworded parts of this paragraph to make is more applicable to the wide target audience.
	line 18-28 (2nd paragraph in introduction)	In response to the reviewers comments, we have reworded parts of this paragraph to make is more applicable to the wide target audience.
	line 29-45 (3rd paragraph in introduction)	We fixed a few grammatical errors and added a few phrases to improve readability.
	line 46-64 (4th paragraph in introduction)	We cut words here to help make room for bringing more of the model description to the main text.
Results: Design of Digitizer	line 65-78 (1st paragraph in Results: Design of Digitizer)	Description of which signal components are analog and digital flowing into and out of the digitizer circuits was added, acronyms have been clarified
	line 79-101 (2nd paragraph in Results: Design of Digitizer)	Condensed phrasing/sentences to fit the word limit of the journal, added more description of why we choose to analyze the top 30% of cells based on feedback from reviewer 1
	line 102-129 (3rd paragraph in Results: Desgin of Digitizer)	Condensed phrasing/sentences to fit the word limit of the journal, added more detail into how the feedforward digitizer topology functions based on reviewer comments to make the paper more accessible to a broad audience
Results: Metrics for Functional Characterization of Digitizer Modules	line 130 -137 (1st paragraph in Results: Metrics for Functional Characterization of Digitizer Modules)	Deleted the original first paragraph in this section to meet journal word limit
	line 138-144 (2nd paragraph in Results: Metrics for Functional Characterization of Digitizer Modules)	Updated text by adding (ROC) and (Box1) references to improve clarity
	line 145 (AUC BOX in Results: Metrics for Functional Characterization of Digitizer Modules)	This was moved to the main text from the sup in response to some of the reviewer comments on knowing how AUC was calculated.
	line 146-156 (3rd paragraph in Results: Metrics for Functional Characterization of Digitizer Modules)	To accommodate moving the AUC description box to the main text we updated the SNR Box reference to Box 2.

	line 157-169 (4th paragraph in Results: Metrics for Functional Characterization of Digitizer Modules)	Updated Sup Figure referencing as mentioned by the reviewers. We also added a few word for improve readability. We also fixed a spelling error.
	line 170-186 (5th paragraph in Results: Metrics for Functional Characterization of Digitizer Modules)	Updated Sup figure referencing as prompted by the reviewers. We also added a couple word to improve readability.
	line 187 (SNR BOX in Results: Metrics for Functional Characterization of Digitizer Modules)	No change
	line 188-210 (6th paragraph in Results: Metrics for Functional Characterization of Digitizer Modules)	Updated Sup figure referencing as prompted by the reviewers. We also added a couple words to improve readability.
	line 211-221 (7th paragraph in Results: Metrics for Functional Characterization of Digitizer Modules)	Updated Sup figure referencing as prompted by the reviewers.
Results: Biophysical Representation of the Governing Mechanics	line 222-237 (1st paragraph in Results: Biophysical Representation of the Governing Mechanics)	We fixed a few grammatical errors
	line 238-249 (2nd paragraph in Results: Biophysical Representation of the Governing Mechanics)	We fixed a few grammatical errors, wrote our a few abbreviations and, in response to reviewer comments, added words to set up our description of the model in the main text in three parts, testing platform, cellular reactions and parameterization.
	line 250-267 (3rd paragraph in Results: Biophysical Representation of the Governing Mechanics)	In response to reviewer comments we have added a paragraph here describing the testing platform.
	line 268-283 (4th paragraph in Results: Biophysical Representation of the Governing Mechanics)	In response to reviewer comments we have added a paragraph here describing the cellular reactions.

	line 284-306 (5th paragraph in Results: Biophysical Representation of the Governing Mechanics)	In response to reviewer comments we have added a paragraph here describing the model parameterization.
	line 307-324 (6th paragraph in Results: Biophysical Representation of the Governing Mechanics)	We fixed grammatical errors as well as added a few more words to improved description of our metric analysis of the model results. Importantly we also updated the reported Pearson correlation data to reflect the true values original found in Sup Table 8.
Results: Creation of Cell- Cell Communication Switch	line 325-333 (1st paragraph in Results: Creation of Cell-Cell Communication Switch)	In response to reviewers comments, added a new paragraph to help introduce the background of synNotch
	line 334-353 (2nd paragraph in Results: Creation of Cell-Cell Communication Switch)	We updated sup table reference and fixed a few grammatical errors
	line 354-367 (3rd paragraph in Results: Creation of Cell-Cell Communication Switch)	We updated sup figure reference and fixed a few grammatical errors
Discussion	line 368-381 (1st paragraph in the Discussion)	No change
	line 382-393 (2nd paragraph in the Discussion)	Minor word changes and added description to make the language accessible to a broader audience
	line 394-403 (3rd paragraph in the Discussion)	Minor word changes and added description to make the language accessible to a broader audience
	line 404-415 (4th paragraph in the Discussion)	No change
	line 416-425 (5th paragraph in the Discussion)	Paragraph added to further detail the applicability of our metrics and modelling techniques based on feedback from all three reviewers
	line 426-438 (6th paragraph in the Disucussion)	No change
Methods	line 439-442 (Intro to Methods)	No change
	line 443-449 (Methods: Cell Maintain ace)	No change
	line 450-474 (Methods: Cell Plating, Transfection and Induction)	We added " and plasmids of similar size to help produced desired ratio of components" as recommended by the reviewers. We also added a brief description and reference to the Sup note discussing our results concerning toxicity in the data, in response to reviewer comments.

	line 475-481 (Methods: Co-Culture Experiments)	No change
	line 482-508 (Methods: FACs Configuration)	No change
	line 509-527 (Methods: TASBE analysis)	As prompted by the checklist we added a reference to a new sup figure describing the processing of our flow cytometry data.
Figures and Remaining Parts of Main Text	Title	No change
	Abstract	No change
	line 528-545 (Data Availability, Conflict of Interest and Author Contributions)	No change
	Line 546-609 (References)	No change
	Figure 1	Updated Figure 1B-1D to include the IFP component of the digitizer circuit
	Figure 2	Updated Figure 2 to include a heat map of fold change in addition to SNR and AUC
	Figure 3	Figure 3C has been updated as while going back to add n values for all our reported Pearson correlations. We found the no-shRNA figure had not been updated to contain all the data used in the study. Therefore, we update this figure. To be complete we also went back a recalculated all values to make sure Sup Table 8 is indeed the most up to date results.
	Figure 4	This figure formatting was updated to display correctly - no major changes were made otherwise
Supplemental	Text	No change
	S Figure 1	This figure has been relabeled as Figure S2. Additionally, error bars representing the SEM of three technical replicates were added to each time point in S2B
	S Figure 2	This figure has been relabeled as Figure S1
	S Figure 3	No change
	S Figure 4	Error bars representing the SEM of three technical replicates were added to each data point in all four plots and indicated them in the caption. Additionally, the legend has been enlarged and the text made more clear to indicate that the different lines are based on discreet CFP-based bins. We identified an error in the code used to plot the 5 ng Flp Plasmid plot

	originally after reading one of Reviewer 1's comment, and have updated the data in the plot.
S Figure 5	Error bars representing the SEM of three technical replicates were added to each data point in Figure S5A
S Figure 6	The text size on the axes of each plot has been increased
S Figure 7	The text size on the axes of each plot has been increased
S Figure 8	Updated caption to indicate error bars represent SEM
S Figure 9	This figure has been updated to remove the partial cut off of the axis label mRuby.
S Figure 10	The size of the legend here has been increased in response to reviewers comments.
S Figure 11	No change
S Figure 12	No change
S Figure 13	Error bars representing the SEM of three technical replicates were added to each data point in Figure S13A and this is noted in the caption.
S Figure 14	No change
S Figure 15	No change
S Figure 16	The text size on axes labels in Figs. S16 B & S16C has been increased. 16C: R ² has been adjusted to r to accurately depict the Pearson Correlation Coefficient
S Figure 17	The text size on axes labels in Figs. S16 B & S16C has been increased. 16C: R ² has been adjusted to r to accurately depict the Pearson Correlation Coefficient
S Figure 18	No change
S Figure 19	No change
S Figure 20	No change
S Figure 21	The text size on axes labels in all panels of Fig S21 has been increased. R ² text has been changed to r to accurately depict the Pearson Correlation Coefficient
S Figure 22	No change
S Figure 23	The text size on the axes labels in Fig S23B-D has been increased
S Figure 24	Caption updated to indicate error bars represent SEM

	S Figure 25	Text in the middle row of Fig S25 has been changed from "Optimal eKd" to "Balanced eKd"
	S Figure 26	Caption updated to indicate error bars represent SEM
	S Figure 27	This figure has been added to detail the exact workflow for gating cells and standardizing flow cytometry measurements to units of GFP MEFL
	S Table 1	No change
	S Table 2	No change
	S Table 3	No change
	S Table 4	No change
	S Table 5	No change
	S Table 6	No change
	S Table 7	We moved this table to the supplement from the main text as suggested by the reviewers.
	S Table 8	We updated the table number as well as added n values for all calculation results displayed in table.

Edits to Main Text Figures – indicated by large red arrows: 
Figure 1: Added IFP

Added depiction of IFP based on reviewer feedback

Figure 2: Added fold change heat map as panel 2C

Figure 3: Updated panel C

Added more data points and updated the r value to the No shRNA plot to represent all data gathered experimentally at this condition

REVIEWERS' COMMENTS:

Reviewer #3 (Remarks to the Author):

I am pleased with the work that the authors have done and with the new additions to the manuscript. The authors have address all my suggestions. The manuscript is more clear and has been improved from the previous version.

I still found a few typos that authors should correct.

146 striaghtforward

254 fluorescent

401-402 dementionalization analsysis,